

# Field intercomparison of prevailing sonic anemometers

Matthias Mauder[1], Matthias J. Zeeman[1]

[1]Karlsruhe Institute of Technology, Institute of Meteorology and Climate Research, Atmospheric Environmental Research, Garmisch-Partenkirchen, Germany

*Correspondence to*: Matthias Mauder (matthias.mauder@kit.edu)

**Abstract.** Three-dimensional sonic anemometers are the core component of eddy-covariance systems, which are widely used for micrometeorological and ecological research. In order to characterize the measurement uncertainty of these instruments we present and analyse the results from a field intercomparison experiment of six commonly used sonic anemometer models

from four major manufacturers. These models include Campbell CSAT3, Gill HS-50 and R3, METEK uSonic-3 Omni, R.M. Young 81000 and 81000RE. The experiment was conducted over a meadow at the TERENO/ICOS site De-Fen in southern Germany over a period of 16 days in June of 2016 as part of the ScaleX campaign. The measurement height was 3 m for all sensors, which were separated by 9 m from each other, each on its own tripod, in order to limit contamination of the turbulence measurements by adjacent structures as much as possible. Moreover, the high-frequency data from all instruments were treated

with the same post-processing algorithm. In this study, we compare the results for various turbulence statistics, which include mean horizontal wind speed, standard deviations of vertical wind velocity and sonic temperature, friction velocity and the buoyancy flux. Quantitative measures of uncertainty, such as bias and comparability, are derived from these results. We find that biases are generally very small for all sensors and all computed variables, except for the sonic temperature measurements of the two Gill sonic anemometers (HS and R3), confirming a known transducer-temperature dependence of the sonic

temperature measurement. The best overall agreement between the different instruments was found for the mean wind speed and the buoyancy flux.

## 1 Introduction

Although sonic anemometers have been used extensively for several decades in micrometeorological and ecological research, there is still some scientific debate about the measurement uncertainty of these instruments. This is due to the fact that an

absolute reference for the measurement of turbulent wind fluctuations in the free atmosphere does not exist. Traditionally, two approaches have been applied to evaluate the performance of sonic anemometers, either by placing them in a wind-tunnel and testing them for different flow angles, or by putting different instruments next to each other in the field over a homogeneous surface, so that all of them can be expected to measure the same wind velocities and turbulence statistics. The first approach has the advantage that the true flow characteristics are well known; however, the characteristics of the flow deviate far from

those in the turbulent atmospheric surface layer where sonic-anemometers are typically deployed. Reynolds numbers in a wind tunnel, for instance, are several orders of magnitude smaller than under natural conditions. In contrast, the second



intercomparison approach has the disadvantage, that it lacks an uncontested reference; however, such field experiments allow the simultaneous evaluation of several instruments under real-world conditions. In other words, the first approach has a high internal validity while the second approach has a high external validity.

Wind-tunnel experiments have been an important milestone towards revealing and quantifying probe-induced flow distortion

effects. One of the first wind-tunnel tests including a correction equation for flow distortion effects is reported by Kaimal (1979). Considering the results of another wind-tunnel study about a three-dimensional hot-wire anemometer, Högström (1982) stressed the importance of such test for all turbulence sensors, and wind-tunnel experiments soon became a standard method for optimizing and calibrating sonic anemometers. Subsequently, Zhang et al. (1986) developed a new sonic anemometer based on measurements of from the wind-tunnel, which inspired the design of the Campbell CSAT3. A further

wind-tunnel calibration for the Gill Solent R2 sonic anemometer is presented by Grelle and Lindroth (1994).

However, researchers soon realized that the transferability of wind-tunnel experiments to field conditions is limited. A very interesting comparative wind-tunnel study about several sonic anemometers (Gill Solent, METEK USA-1, Kaijo Denki TR-61A, TR-61B, and TR-61C) is conducted by Wieser et al. (2001). They evaluate flow distortion correction algorithms provided by the respective manufacturers and come to the following conclusion „Because of the very low level of turbulence in the wind

tunnel (no fences or trip devices have been used), the size and stability of vortices set up behind struts may be increased in comparison with field measurements." (Wieser et al., 2001) Moreover, Högström and Smedman (2004) present a critical assessment of laminar wind-tunnel calibrations by using a hot-film instrument as reference during a field experiment over a flat and level coastal area with very low vegetation. Their results indicate that wind-tunnel based corrections might be overcorrecting, or at least do not improve the comparison with the reference measurement of turbulence statistics.

Despite these known limitations, more extensive wind-tunnel calibration studies were conducted, which led to the publication of the so-called angle-of-attack correction for Gill Solent R2 and R3 (van der Molen et al., 2004; Nakai et al., 2006). However, it is often overlooked that angle-of-attack dependent errors might partially be an artefact of wind-tunnel experiments, because in quasi-laminar wind-tunnel flows the angle-of-attack remains constant. In contrast, the flow distorting caused by the same geometrical structure is much smaller under turbulent conditions, when the three-dimensional wind vector and the

corresponding flow angles fluctuate constantly (Huq et al., 2017).

In order to address concerns about the validity of these wind-tunnel based calibrations, the angle-of-attack based flow distortion concept was investigated in the field under natural turbulent conditions. Nakai and Shimoyama (2012) mounted several Gill WindMaster instruments at different angles next to each other above a short grass canopy, and Kochendorfer et al. (2012) conducted a very similar field experiment focussing on RM Young Model 81000 anemometers, while the Campbell CSAT3

was only briefly examined. It has to be noted that the results of these two studies were interpreted under the false assumption that the instantaneous wind vector remains unchanged between different instruments that are mounted more than 1 m apart, which contradicts the concept of a fluctuating turbulent flow with a certain decay of the spatial autocorrelation function (Kochendorfer et al., 2013; Mauder, 2013).





However, such side-by-side comparisons with different alignment of the same instrument can be quite instructive, as long as only turbulence statistics are analysed, which can indeed be considered to be similar across several metres over homogeneous surfaces. Although their study site is less than ideal for a field intercomparison (over a sloped forest canopy within the roughness sublayer), Frank et al. (2013) found that non-orthogonal positioned transducers can underestimate vertical wind velocity (w) and sensible heat flux (H), by comparing the output of two pairs of CSAT3 anemometers while one pair was rotated by 90°. This finding was substantiated in a follow-up study (Frank et al., 2016), which also covers a side-by-side comparison of two CSAT3 mounted at different alignment angles plus two sonic anemometers with an orthogonal transducer array and a CSAT3 with one vertical path. An elaborate statistical analysis leads them to the conclusion: "Though we do not know the exact functional form of the shadow correction, we determined that the magnitude of the correction is probably somewhere between the Kaimal and double-Kaimal correction." (Frank et al., 2016), referring to the original work of Kaimal (1979).

In a parallel chain of events, International Turbulence Comparison Experiments (ITCE) were carried out at different places around the world since the early days of sonic anemometry used for micrometeorological field campaigns (Dyer et al., 1982; Miyake et al., 1971; Tsvang et al., 1973, 1985), mostly with the aim to investigate the comparability of different instrumental designs. Typically, relative differences were analysed based on those comparative datasets, which generally suffer from the lack of a "true" reference measurement or etalon, but those experiments have the advantage that many anemometer models can be tested at once under real-world conditions. Nevertheless, also absolute biases were sometimes detected, such as the flow distortion from supporting structures, which from the 1976 ITCE was deduced from a non-zero mean vertical wind speed, especially for geometries with a supporting rod directly underneath the measurement volume (Dyer, 1981).

In those early ITCEs, mostly custom-made instruments were tested. However, since the beginning of the 1990s, a growing number of commercial sonic anemometer models became available from a number of manufacturers. Based on their field intercomparison experiments, Foken and Oncley (1995) classified all instruments commonly used at the time according to their expected errors into those that are suitable for fundamental turbulence research and those that are sufficient for general flux measurements. About one decade later, several then popular models were compared in a thorough and comprehensive study by Loescher et al. (2005). They tested eight different probes for the accuracy of their temperature measurement in a climate chamber; they investigated biases of the w-measurement in a low-speed wind tunnel, and investigated differences in the turbulence statistics measured in the field. At about the same time, also Mauder et al. (2007) conducted a field intercomparison of seven different sonic anemometers as part of the international energy balance closure experiment EBEX-2000 above a cotton field in California. Both studies more or less confirmed the classification of Foken and Oncley (1995) who concluded that only the directional probes without supporting structure directly underneath the measurement volume meet the highest requirements of turbulence research, while no significant deviations between those top-class instruments were detected.

The persisting lack of energy balance closure at many sites around the world (Stoy et al., 2013) and the emerging indications a general flux underestimation of non-orthogonal sonic arrays (Frank et al., 2013) are the primary motivation of a special field



experiment by Horst et al. (2015). They conducted an intercomparison at an almost ideal site, which was flat, even and with a homogeneous fetch. Two CSAT3 representing a typical non-orthogonal sensor, were compared against two different orthogonal probes manufactured by Applied Technologies Inc. and one custom-made CSAT3 with one vertical path. Under the assumption that the flow-distortion correction of Kaimal (1979) is correct, they state that the CSAT3 requires a correction

of 3% to 5%. This is in quite good agreement with the conclusion of Frank et al. (2016), who suggest a correction of the magnitude between Kaimal and double Kaimal, and the numerical study of Huq et al. (2017), which found an underestimation of 3% to 7%. Thus, at least for the CSAT3, some consensus is emerging about the magnitude of the correction required under turbulent conditions in the field.

Although the results on measurement error are converging for the CSAT3 model, less is known about the comparability

between different sonic anemometer models available today. As the last comprehensive intercomparison experiments were conducted more than 10 years ago, and some new models have emerged on the market since then and some others have received firmware upgrades, we believe it is time for another field intercomparison covering commonly used sonic anemometers. We deployed six different models from four different manufacturers next to each other over a short grass canopy. Furthermore, two CSAT3 were tested simultaneously in order to compare the influence of transducer rainguards. An orthogonal regression

analysis is applied to the turbulence statistics obtained from the different instruments, and quantitative measures of uncertainty, such as bias and comparability (RMSE), are derived.

## 2 Materials and methods

### 2.1 Field experiment

This sonic anemometer inter-comparison experiment took place at the Fendt field site in Southern Germany (DE-Fen,

47.8329°N 11.0607°E, 595 m a.s.l.), which belongs to the German Terrestrial Environmental Observatories (TERENO) network. The measurement period was from 06 June to 22 June 2016, and the intercomparison was conducted as part of the multi-scale field campaign ScaleX (Wolf et al., 2017), where the sonic anemometers were subsequently deployed at different locations. The landscape surrounding the site comprises gentle hills that are partially covered by forest (Figure 1), and the land cover within the footprint consisted of grassland with a canopy height of 0.25 m (Zeeman et al., 2017). The aerodynamic

roughness length was estimated to be 0.03 m. In this field experiment, we compared seven sonic anemometers from four different manufacturers. A detailed list of all participating instruments is provided in Table 1.

Since the dominant wind direction is North for this site on typical summer days due to a thermal circulation between the Alps and the Alpine foreland (Lugauer and Winkler, 2005), we set-up all instrumented towers in a row from East to West. The sensors were separated by 9 m from each other in order to avoid flow distortion between neighboring towers. The measurement

height of all sonic anemometers was 3.0 m, and they were oriented towards West (270°) for all non-omnidirectional probes (Figure 2). Data from all instruments were digitally recorded on synchronized single-board computers (BeagleBone Black,



BeagleBoard.org Foundation, Oakland Twp, MI, USA), using an event-driven communication protocol implemented in Python programming language.

Figure 1: Location of the sonic anemometer (SA) transect at the DE-Fendt field. Map modified from Fig 1 in Zeeman et al. (2017)



**Figure 2: The bottom part of this figure shows a photograph of the field intercomparison experiment; the micrometeorological installations of the TERENO/ICOS site DE-Fen can be seen in the background (left). On top, close-up pictures of all seven sonic anemometers are shown.**

Figure 3 shows the meteorological conditions during the experiment. As expected for this site and for this time of the year, the dominant daytime wind direction was North. Wind speeds ranged between 0 and 5 m s$^{-1}$. Air temperatures varied between 8° C and 24° C. Net radiation reached values up to 700 W m$^{-2}$. On 08, 09, 19 June, the cloud cover was rather dense all day. Most of the days are characterized be high loads of net radiation with values larger than 500 W m$^{-2}$ at maximum. Nevertheless, also

10   rain occurred on most of the days with the exception of first two days of the measurement period, 06 - 07 June, and the last day, 22 June. Overall, this experiment can be considered as being typical conditions in the early summer of temperate climate zones.





**Table 1: Participating instruments in the order of their location from East to West**

| Comparison name | CSAT3_1 | Gill.R3 | Gill.HS | Metek.uSonic3.omni | Young.81000 | Young.81000RE | CSAT3_2 |
|---|---|---|---|---|---|---|---|
| Manufacturer | Campbell Scientific Inc. | Gill Instruments Ltd. | Gill Instruments Ltd. | METEK Meteorologische Messtechnik GmbH | R. M. Young Company | R. M. Young Company | Campbell Scientific Inc. |
| Model | CSAT3 | 1210R3 | HS | uSonic-3 Omni AH | 81000 | 81000RE | CSAT3 |
| Serial number | 1791 | 585 | 152903 | 0106054006 | 003149 | UA 02043 | 0771 |
| Path length (mm) | 116 | 150 | 150 | 138 | 150 | 150 | 116 |
| Transducer diameter (mm) | 6.4 | 11 | 11 | 13.8 | 13.8 | 13.8 | 6.4 |
| Transducer path angle (°) | 60 | 45 | 45 | 45 | 45 | 45 | 60 |

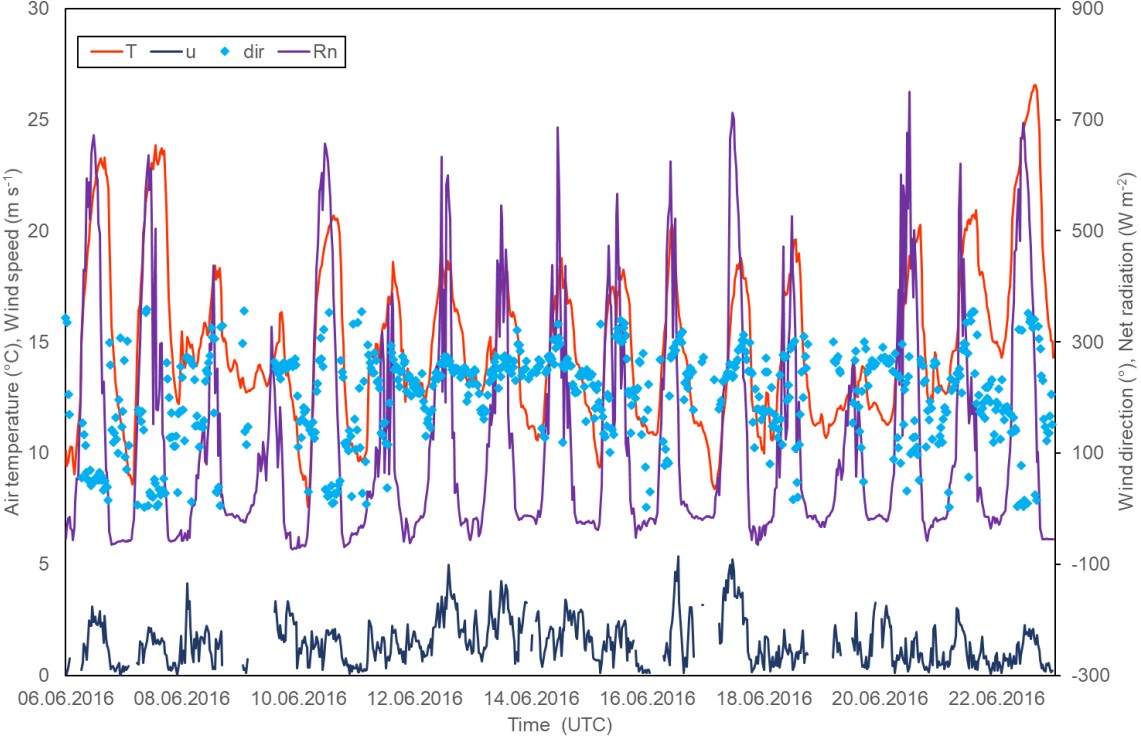

**Figure 3: Meteorological elements during the intercomparison experiment; 30-min averages of air temperature (*T*) and net radiation (*Rn*) measured at 2 m, and wind speed (*u*) and direction (*dir*) from the DE-Fen site measured at 3.25 m.**



## 2.2 Data processing

All data were processing using the TK3 software (Mauder and Foken, 2015) according to the processing scheme of Mauder et al. (2013). More precisely, turbulent statistics were calculated using 30-min block averaging, after applying a spike removal algorithm on the high-frequency raw data. We applied the double rotation method (Kaimal and Finnigan, 1994) and a spectral

correction for path averaging according to Moore (Moore, 1986). The compared turbulent quantities are defined as follows:

$U = \bar{u}$, the averaged total wind velocity after alignment of the coordinate system into the mean wind;

$T_s = \bar{T_s}$, the averaged sonic temperature;

$\sigma_w = \sqrt{\overline{w'w'}}$, the standard deviation of the vertical velocity component;

$\sigma_{Ts} = \sqrt{\overline{T_s'T_s'}}$, the standard deviation of the sonic temperature;

$u_* = \sqrt[4]{\overline{u'w'}^2 + \overline{v'w'}^2}$, the friction velocity calculated from both covariances between the two horizontal wind components and $w$;

$H_s = \rho c_p \overline{w'T_s'}$, the buoyancy flux calculated from the air density $\rho$, the specific heat capacity at constant pressure $c_p$ and the covariance between $w$ and $T_s$.

These quantities were filtered for rain (during the respective half hour or the half hour before as recorded by a Vaisala WXT520

sensor of the nearby TERENO station DEFen), obstructed wind directions $\varphi$ ($70° < \varphi < 110°$; $250° < \varphi < 290°$) and non-steady-state conditions, i.e. data with Foken et al. (2004) steady state test flag 4-9, considering the $u_*$-flag for all statistics concerning the pure wind measurements ($U$, $\sigma_w$, $u_*$) and the sensible heat flux-flag for all statistics that include sonic temperature ($T_s$, $\sigma_{Ts}$, $H_s$).

The reference instrument (etalon) was chosen for each compared quantity independently according a principle component

analysis (PCA) using the R function princomp(). We selected the instrument with the highest loading on the first principle component. Only when the Young.81000RE had received the highest loading, we selected the sonic anemometer with the second highest loading as etalon instead because the Young.81000RE time series only starts more than three days later at 10.06.2016 14:00 due to technical issues in the beginning of the field experiment.

For the statistical analysis of the intercomparison, an orthogonal Deming regression was applied in order to account for

measurement errors in both x and y variables, using the R package mcr (Manuilova, E., Schuetzenmeister and Model, 2014). Furthermore, we calculated the values for comparability, which is equivalent to the root-mean-square-error (RMSE), and bias, which is the mean error for a certain measurement quantity.





## 3 Results

### 3.1 Mean wind speed

For our comparison of the mean wind velocity measurements, the METEK.uSonic3.omni was selected as etalon, because it received the highest loading (-0.3785) on the first principle component of our PCA. However, the loadings of the two Gill

5 instruments and the YOUNG.81000 are not much lower either. Hence, the two Gill anemometers and the Young.81000 compare slightly better with the etalon than the rest. Nevertheless, the agreement between of the $U$-measurements by all tested anemometers is generally very good, as can be seen from Figure 4. This is also indicated by small regression intercepts (< 0.04 m s$^{-1}$) and slopes close to one (1±0.03). In general, comparability values are smaller than 0.11 m s$^{-1}$ and biases range between −0.05 m s$^{-1}$ and 0.06 m s$^{-1}$ (Table 2). The agreement between the two CSAT3 is as good as the overall agreement

10 between all tested instruments.

**Table 2: Regression results for the comparison of mean wind velocity $U$, plus estimates for bias and comparability (RMSE).**

| etalon = METEK.uSonic3.omni | CSAT3_1 | Gill.R3 | Gill.HS | Young.81000 | Young.81000RE | CSAT3_2 |
|---|---|---|---|---|---|---|
| n | 367 | 367 | 367 | 366 | 257 | 367 |
| intercept (m s$^{-1}$) | -0.01 | 0.00 | 0.01 | 0.03 | 0.00 | 0.04 |
| slope | 1.02 | 1.02 | 1.03 | 1.02 | 0.97 | 0.97 |
| bias (m s$^{-1}$) | 0.02 | 0.04 | 0.05 | 0.06 | -0.05 | 0.00 |
| RMSE (m s$^{-1}$) | 0.11 | 0.08 | 0.07 | 0.08 | 0.11 | 0.09 |






**Figure 4: Comparison of the 30 min averaged wind velocity measurements (etalon = METEK.uSonic.omni).**

## 3.2 Mean sonic temperatures

The ultrasound-based temperature measurement is determined from the absolute time of flight as opposed to the differences
in time flight for the velocity measurement. Therefore, inaccuracies in pathlength due to inadvertent bending or varying
electronic delays of the signal processing directly affect the accuracy of the measurement, and it is not surprising that the
general agreement between different instruments is much worse for the sonic temperature than for the wind velocity. The
Young.81000 received the highest loading (-0.3806) and was therefore chosen as etalon. Good agreement with this reference
is found for the two CSAT3 and the METEK.uSonic.omni, which is indicated by values well below 1 K for bias and
comparability. However, larger discrepancies occur for the two Gill sonic anemometers and the Young.81000RE. As can be
seen from Figure 5, the Young.81000RE sonic temperatures show a linear relationship with the etalon, so that the error of this
instrument could be corrected by a simple regression equation using the coefficients provided in Table 3. In contrast, the sonic
temperature measurements of the two Gill sensors show much more scatter and non-linearity in addition to a large bias, which



is determined as 1.82 K for the Gill.R3 and 3.55 K for the Gill.HS. Therefore, the comparability values are also large with RMSE = 1.99 K for the Gill.R3 and 3.58 K for the Gill.HS.

![Figure 5: Six scatter plots comparing sonic temperature measurements for CSAT3_1, Gill.R3, Gill.HS, METEK.uSonic3.omni, Young.81000RE, and CSAT3_2.]

**Figure 5: Comparison of the averaged sonic temperature measurements (etalon = Young.81000).**

5    **Table 3: Regression results for the comparison of mean sonic temperature $T_s$, plus estimates for bias and comparability (RMSE); unusually large deviations from the etalon are underlined.**

| etalon = Young.81000 | CSAT3_1 | Gill.R3 | Gill.HS | METEK.uSonic3.omni | Young.81000RE | CSAT3_2 |
|---|---|---|---|---|---|---|
| n | 321 | 321 | 321 | 321 | 229 | 321 |
| intercept (K) | 0.01 | 2.22 | 3.37 | -0.18 | 0.69 | -0.15 |
| slope | 1.05 | 0.97 | 1.01 | 1.05 | 1.06 | 1.04 |
| bias (K) | 0.75 | 1.82 | 3.55 | 0.58 | 1.69 | 0.49 |
| RMSE (K) | 0.79 | 1.99 | 3.58 | 0.62 | 1.70 | 0.54 |



### 3.3 Standard deviation of the vertical velocity component

An accurate and precise measurement of the standard deviation of the vertical velocity component is particularly important because the *w*-fluctuations are required for the determination of any scalar flux by eddy covariance, also those fluxes that require the deployment of an additional sensor, such as an infrared gas analyser or other laser-based fast-response sensors.

5  During our field experiment, $\sigma_w$ values ranged between 0 m s$^{-1}$ and 0.7 m s$^{-1}$. The Gill.HS anemometer was chosen as for $\sigma_w$ as it received the highest loading from our PCA (-0.3781). All other instruments agree very well with this reference as can be seen from Figure 6. Intercepts and biases are very small, ranging from −0.01 m s$^{-1}$ to 0.02 m s$^{-1}$ (Table 4). Values for comparability are better than 0.02 m s$^{-1}$ and the regression slopes are close to one (1±0.03).

**Figure 6: Comparison of the standard deviation of the vertical velocity component $\sigma_w$ (etalon = Gill.HS)**





**Table 4: Regression results for the comparison of the standard deviation $\sigma_w$, plus estimates for bias and comparability (RMSE).**

| etalon = Gill.HS | CSAT3_1 | Gill.R3 | METEK.uSonic3.omni | Young.81000 | Young.81000RE | CSAT3_2 |
|---|---|---|---|---|---|---|
| n | 367 | 367 | 367 | 366 | 257 | 367 |
| intercept (m s$^{-1}$) | 0.00 | 0.00 | 0.00 | 0.02 | -0.01 | -0.01 |
| slope | 0.98 | 1.01 | 0.97 | 0.99 | 0.98 | 1.03 |
| bias (m s$^{-1}$) | -0.01 | 0.00 | 0.00 | 0.01 | -0.01 | 0.00 |
| RMSE (m s$^{-1}$) | 0.02 | 0.01 | 0.02 | 0.02 | 0.02 | 0.02 |

## 3.4 Standard deviation of the sonic temperature

Despite the large discrepancies of the mean sonic temperature measurements between certain instruments, the fluctuations of sonic temperature agree much better (Figure 7). For this turbulent quantity, the CSAT2_2 was chosen as etalon although it only had the second-highest loading in our PCA (-0.3816) because the Young.81000RE, which received a slightly higher loading (-0.3824), only recorded data four days after the comparison experiment had begun. None of the tested instruments shows a large bias nor a large regression intercept for the measurement of $\sigma_{Ts}$. However, the large errors in mean sonic temperature of the two Gill anemometers also lead to a larger scatter for $\sigma_{Ts}$, which expresses itself in comparability values larger than 0.06 K for the Gill.HS and 0.08 K for the Gill.R3 (Table 5). Surprisingly, the Young.81000 has an even poorer comparability of 0.09 K – it was the etalon for the mean sonic temperature measurement. In contrast, the Young.81000RE shows a very good agreement with the etalon for $\sigma_{Ts}$ despite its large bias when measuring mean sonic temperature. The METEK.uSonic.omni stands out because it has the highest regression slope of 1.06, which might be a direct consequence of the almost equally high regression slope of 1.05 for the mean sonic temperature measurement. The agreement between the two CSAT3 is except for a few outliers, which were not rejected by our data-screening algorithm.

**Table 5: Regression results for the comparison of the standard deviation $\sigma_{Ts}$, plus estimates for bias and comparability (RMSE); unusually large deviations from the etalon are underlined.**

| etalon = CSAT3_2 | CSAT3_1 | Gill.R3 | Gill.HS | METEK.uSonic3.omni | Young.81000 | Young.81000RE |
|---|---|---|---|---|---|---|
| n | 322 | 322 | 322 | 322 | 321 | 229 |
| intercept | 0.00 | 0.01 | 0.01 | -0.01 | 0.02 | 0.00 |
| slope | 1.01 | 0.99 | 0.96 | 1.06 | 1.05 | 1.04 |
| bias (K) | 0.00 | 0.00 | -0.01 | 0.01 | 0.04 | 0.01 |
| RMSE (K) | 0.05 | 0.08 | 0.06 | 0.05 | 0.09 | 0.03 |



**Figure 7: Comparison of the standard deviation of the sonic temperature $\sigma_{Ts}$ (etalon = CSAT3_2).**

## 3.5 Friction velocity

5   Friction velocities ranged between 0 m s$^{-1}$ and almost 0.6 m s$^{-1}$ during our experiment. Although the Yount.81000RE has the highest loading (-0.3803) in our PCA, we chose the Gill.HS as etalon due to the abovementioned data-availability issue of the Young.81000RE, but again its loading is only slightly lower (-0.3801). For $u_*$, generally much larger scatter is observed than for other purely wind-related quantities, such as $U$ and $\sigma_w$ (Figure 8), which manifests itself in comparability values of 0.05 m s$^{-1}$ or 0.06 m s$^{-1}$ respectively (Table 6). However, despite the large scatter the biases and regression intercepts are

10   generally smaller with values lower than 0.02 m s$^{-1}$ in absolute numbers. Only the METEK.uSonic.omni measures friction velocities consistently larger than the etalon on average, which manifests itself in a bias and regression intercept of 0.03 m s$^{-1}$.



The relatively low regression slope of the CSAT3_1 of 0.91 does not lead to unusually poor error estimates of neither comparability (0.05 m s$^{-1}$) nor bias (−0.01 m s$^{-1}$).

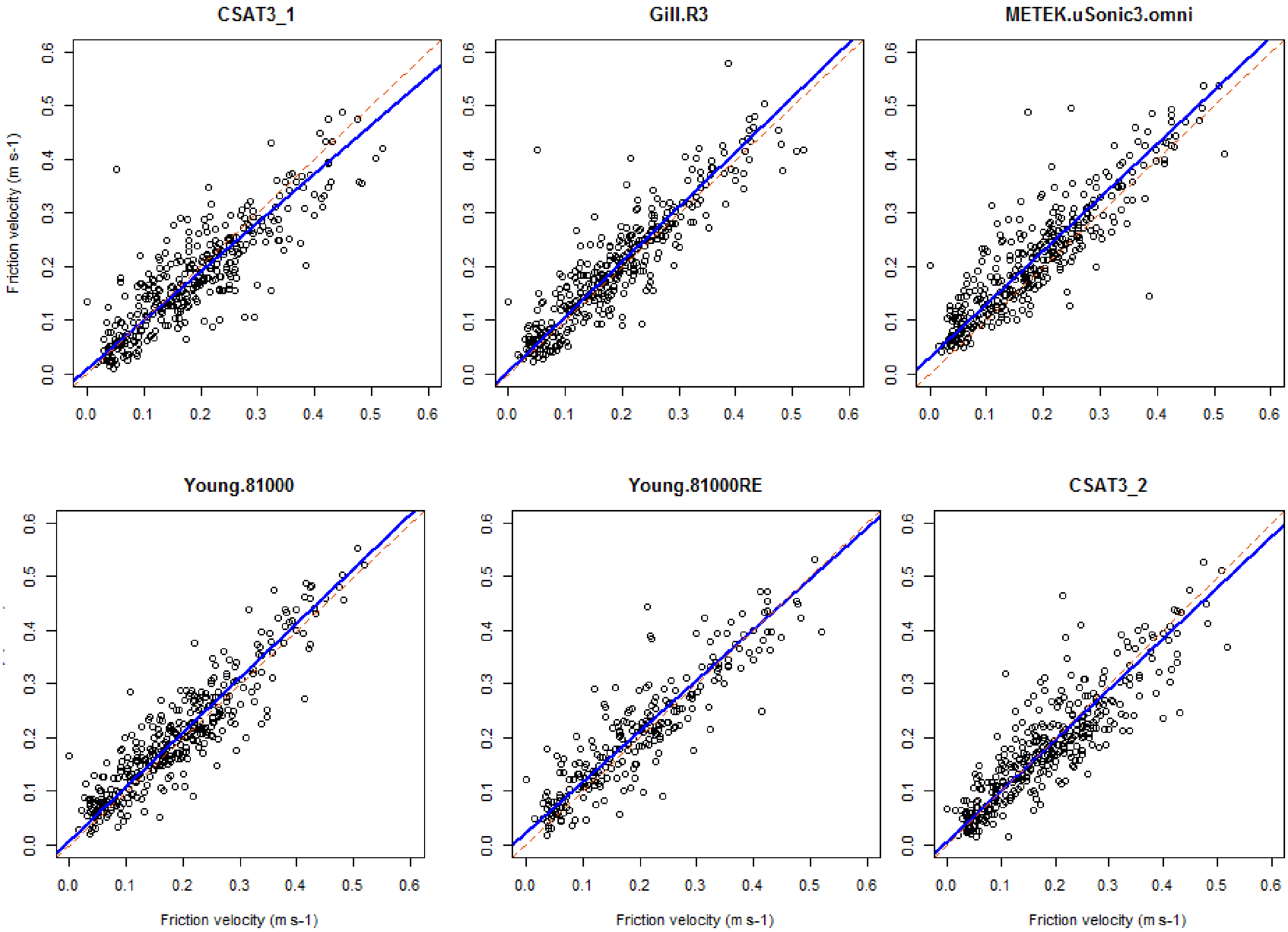

**Figure 8: Comparison of the friction velocity measurements (etalon= Gill.HS).**

**Table 6: Regression results for the comparison of friction velocity $u_*$, plus estimates for bias and comparability (RMSE); unusually large deviations from the etalon are underlined.**

| etalon = Gill.HS | CSAT3_1 | Gill.R3 | METEK.uSonic3.omni | Young.81000 | Young.81000RE | CSAT3_2 |
|---|---|---|---|---|---|---|
| n | 365 | 362 | 365 | 364 | 255 | 364 |
| intercept (m s$^{-1}$) | 0.01 | 0.00 | 0.03 | 0.01 | 0.02 | 0.01 |
| slope | <u>0.91</u> | 1.03 | 1.00 | 1.02 | 0.95 | 0.95 |
| bias (m s$^{-1}$) | -0.01 | 0.01 | 0.03 | 0.01 | 0.01 | 0.00 |
| RMSE (m s$^{-1}$) | 0.05 | 0.05 | 0.06 | 0.05 | 0.05 | 0.05 |





## 3.6 Buoyancy flux

Quantifying fluxes by eddy covariance is probably the most common application of sonic anemometers. Therefore, the comparison of the buoyancy flux measurements is perhaps the most interesting aspect of this study from many researchers. At

5  first, we would like to note that the number of available data is reduced by about one third compared to the other quantities, which is due to rejection of instationary periods by the quality tests of Foken et al. (2004). For this comparison, the CSAT3_1 was chosen as etalon for this quantity because it received the highest loading in our PCA (-0.3786). The overall agreement between all sonic anemometers is excellent as can be seen from Figure 9. Biases are generally very small with values less than 3 W m$^{-2}$, and all of the regression slopes are very close to one (1±0.02) (Table 7). Some minor scatter that is apparent in the

10  comparison plots of Figure 9 results in comparability values between 8.6 W m$^{-2}$ and 11.2 W m$^{-2}$ for the different instruments.

**Figure 9: Comparison of buoyancy flux measurements (etalon = CSAT3_1)**



**Table 7: Regression results for the comparison of buoyancy flux $H_s$, plus estimates for bias and comparability (RMSE)**

| etalon = CSAT3_1 | Gill.R3 | Gill.HS | METEK.uSonic3.omni | Young.81000 | Young.81000RE | CSAT3_2 |
|---|---|---|---|---|---|---|
| n | 219 | 224 | 209 | 210 | 153 | 211 |
| intercept (W m$^{-2}$) | 0.0 | 1.2 | 0.9 | -2.5 | 0.8 | 0.7 |
| slope | 1.02 | 1.00 | 1.02 | 1.00 | 0.98 | 0.98 |
| bias (W m$^{-2}$) | 0.7 | 1.4 | 1.7 | -2.6 | 0.0 | -0.3 |
| RMSE (W m$^{-2}$) | 9.4 | 8.6 | 11.2 | 10.5 | 8.6 | 10.0 |

**4 Discussion**

In theory, the overall agreement between sonic anemometers cannot be better than the random error, if the seven different measurements systems collect independent samples of an homogeneous turbulence field (Richardson et al., 2012). The stochastic error of due to limited sampling of the turbulent ensemble (Finkelstein and Sims, 2001) is 17% or 0.03 m s$^{-1}$ on average for $u_*$ and 14% or 5 W m$^{-2}$ for $H_s$, based on data from CSAT3_1. The comparability values that we found between different instruments for these two quantities are only slightly larger. This means, a better agreement is hardly physically

possible, and the remaining small discrepancies can be explained by slight surface heterogeneities within the footprint area of the different systems and by a very small instrumental error. The agreement between the two CSAT3 was as good as the agreement with other sonic anemometer models. The rainguards on the CSAT3 unit with serial number 1791 had no visible influence on the measurement performance in comparison to the unit with serial number 0771, and the number of available half-hour statistics was not affected either.

We found a much better agreement between different sonic anemometers, especially for $u_*$ and $H_s$, in comparison to previous intercomparison experiments (Loescher et al., 2005; Mauder et al., 2007). Perhaps this can partially be explained by a consistent digital data acquisition, implemented here with a very high precision clock and event-driven communication using Python programming language. Probably, the implementation of a more efficient spike removal algorithm for the high-frequency data and other additional quality tests in the post-processing scheme of Mauder et al. (2013) also helped to improve

the data quality of the resulting fluxes and consequently improved the agreement. A contribution by changes in the firmware of the different sonic anemometers over the last ten years are likely but not fully documented.

Especially considering flow distortion errors on the order of 5% or more that are reported in the literature (Frank et al., 2016; Horst et al., 2015; Huq et al., 2017), the very good agreement between all sonic anemometers in this field experiment is somewhat surprising. According to the manufacturer, two CSAT3 sonic anemometers have no flow distortion correction at all,

while all the other five instruments probably do apply some sort of correction, only the exact details are not known for all of them. This could mean that flow distortion errors are indeed significant for our experiment but perhaps all instruments are





afflicted with an error of almost the exact same magnitude and consequently underestimate $\sigma_w$ and vertical scalar fluxes similarly, despite the obvious differences in sensor geometry and internal data processing.

Alternatively, one might also suppose that the flow distortion errors were generally small for our experimental set-up due to the occurred distribution of instantaneous flow angles, since flow-distortion effects tend to be smaller for smaller angles of

attack as indicated by the studies of Grelle and Lindroth (1994) and Gash and Dolman (2003). However, the standard deviation of the angles-of-attack was about 15°, which is comparable to other field experiments. For comparison, Gash and Dolman (2003) report about 90% of their data to be within ±20° for the Horstermeer peat bog site, and Grare et al. (2016) report their data to be in a range of ±15°, most of times even within ±10°, measuring at 10 m above shrubland. Horst et al. (2015) report their angles-of-attack to be mostly within ±8° for measurements above low weeds and crop stubble with an aerodynamic

roughness length of 0.02 m. Since the spread of angles of attack is on the upper end of the values reported in the literature, our comparison results can be considered as a conservative estimate for the random instrument-related uncertainty of typical applications of eddy-covariance measurements over vegetation canopies. A common significant systematic error of all tested instruments might nevertheless be possible, as suggested by Frank et al. (2016).

One exception from the overall very good agreement is the sonic temperature measurement by both Gill sonic anemometers,

the HS and the R3. This error appears not only as an offset, but also as deviation of a linear functional relationship and increased scatter. A similar behaviour of other Gill anemometers has been reported before, and also a possibly explanation has been provided in the past (Mauder et al., 2007; Vogt, 1995). Obviously, the sonic temperature measurement of Gill anemometers is compromised by a temperature-dependence of the transducer delay, i.e. the time delay between the arrival of a sound pulse at the transducer and the registration by the electronics board.

**5 Conclusions**

Generally, biases and regression intercepts were very small for all sensors and all computed variables, except for the temperature measurements of the two Gill sonic anemometers (HS and R3), which are known to have a transducer-temperature dependence of the sonic temperature measurement (Mauder et al., 2007). Nevertheless, the Gill anemometers show an equally good agreement for other turbulence statistics. The comparability (RMSE) of the instruments is not always as good as the bias,

indicating a random error that is slightly larger than any systematic discrepancies. The best overall agreement between the different instruments was found for the quantities $U$, $\sigma_w$, and $H_s$, which suggests that the sensors' physical structure and internal signal processing are designed for measuring wind speed and vertical scalar fluxes as accurately as possible. However, the relative random uncertainty of $u_*$ measurements is still large, pointing at the particular challenge in measuring the covariance of horizontal and vertical wind components due to the rather small spectral overlap.

The uncertainty estimate of Mauder et al. (2006) for the buoyancy flux measurement of 5% or 10 W m$^{-2}$ was confirmed, not only for those instruments that were classified in that study as "type A" (CSAT3 and Gill HS), but also for those that were labelled "type B" (Gill R3) back then and all other tested instruments (METEK uSonic3-omni, RM Young 81000 and





81000RE). Hence, from our results we cannot derive a classification of the tested sonic anemometers in different quality levels, which means that the evolution of anemometers by all major manufacturers has converged over the last decade.

For applications aiming at measuring vertical scalar fluxes, all tested instruments can be considered equally suitable, at least for low vegetation ecosystems, as long as digital data acquisition is implemented to avoid additional uncertainty and a stringent
data quality control procedure is applied to detect malfunction of the eddy-covariance system. Moreover, the deviations between instruments of different manufacturers are not larger than between different serial numbers of the same model. Therefore, we do not consider it to be necessary to agree on one single anemometer model to ensure comparability, e.g. for intensive field campaigns or for networks of ecosystem observatories. Instead, other criteria should be taken into account for the selection of a sonic anemometer, such as climatic conditions of a measurement site (e.g. frost, fog, heat), the distribution
of wind directions (omnidirectional or not), the measurement height (path length), the compatibility with an existing data acquisition system or a certain scientific objective. In principle, this conclusion is not in contradiction with the classification Foken and Oncley (1995) and Mauder et al. (2006), because they also concluded that all instruments under investigation were suitable for general flux measurements. Only for specific questions of fundamental turbulence research, it was advised to use specific types of instruments.

Although a good agreement between six different sonic anemometer models indicates a high precision of these type of instruments in general, a field intercomparison study can only provide limited insights on the absolute accuracy of these measurements. Particularly, a systematic error that is common to all tested instruments can inherently never be detected in this way. In the past, wind-tunnel experiments were conducted for this purpose, although their transferability to real-world conditions was always debated. Numerical simulations of probe-induced flow distortion (Huq et al., 2017) may provide a better
way to characterize the suitability of sonic anemometers for turbulence measurements in the future. If systematic errors for one certain instrument are known from these computationally very expensive simulations, then classical field inter-comparisons can be used to test models against such a well-characterized sensor. Moreover, a comparison with a remote sensing based system that is free of flow-distortion, such as LiDAR, would be very helpful if it is able to sample a similarly small volume of air at a similar measurement rate as a sonic anemometer.


**Acknowledgements**

We are grateful to Kevin Wolz and Peter Brugger for field assistance and to Sandra Genzel for help with the data preparation. We thank GWU-Umwelttechnik GmbH for generously lending us the Young.81000RE instrument for the duration of the field experiment. The Fendt site is part of the TERENO and ICOS-D ecosystems (Integrated Carbon Observation System, Germany)
networks which are funded, in part, by the German Helmholtz Association and the German Federal Ministry of Education and Research (BMBF). We thank the Scientific Team of ScaleX Campaign 2016 for their contribution. This work was conducted within the Helmholtz Young Investigator Group "Capturing all relevant scales of biosphere-atmosphere exchange – the enigmatic energy balance closure problem", which is funded by the Helmholtz-Association through the President's Initiative and Networking Fund.





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
