# Peer review of "Field intercomparison of prevailing sonic anemometers"

_Atmospheric Measurement Techniques, 2017_

## Referee Comment (RC2)

Field intercomparison of prevailing sonic anemometers

Matthias Mauder, Matthias J. Zeeman

**General Comments**

"As the last comprehensive intercomparison experiments were conducted more than 10 years ago, …" is the motivation for the authors do carry out a new intercomparison for prevailing sonic anemometers. They present the analysis and the results in a well-prepared manuscript in a straight and standard way. The sonic-user community will be eager to see how the different instrument types perform. Insofar it is worthwhile to publish their results and it is perfectly within the scope of AMT. There are however some points the authors should address.

The explanation/discussion of the much better agreement is not convincing. I don't understand what the consistent digital data acquisition has to do with the better agreement. Give an example. And demonstrate how your quality tests improve the agreement. You also indicate that contributions in changes of firmware might have an influence. Say more about that. And finally you say that five instruments apply "some sort" of correction. There is more information about the corrections, they should not be treated as blackbox. Add the information where it is existiong. E.g. the calibration files for R3 and HS are available and can be applied later (at least it used to be like that). One has the possibility to sample uncalibrated data and apply the calibration afterwards. Did you do that? The HS and the CSAT3: how would they compare then? The sonics from Young and Metek seem to be black boxes but they allow to switch on and off a wake or head correction. You probably used the sonics always with the corrections on. Any idea how strong the corrections are? It irritates me that an instrument like the Young 81000 with a magic wake correction is so close to the other instruments. Insofar I can understand that you find the good agreement "somewhat surprising" and I can follow your conclusion that this is rather a conservative estimate because of special conditions (small variation in angles-of-attac. I guess you investigated the differences on azimuthal dependencies).

The angle-of-attack figure disappeared. It was surprising to see that the deviations from horizontal were that small (mostly within ±6°). Now there is a standard deviation of 15°. What happened?

I do not see the advantage of using the PCA load in the first place, for deciding on an etalon. Choosing rather one instrument for all comparison is much more stringent and makes it easier to compare the instruments.

Comparison plots are a bit monotone and do not transmit much information. Plotting rather differences to reference than sonic value versus sonic value gives an immediate impression on statistics. For a direct connection to the scatter plots the regression results should be placed in the plots. Special features can be highlighted in the text.

Comment on the speed-related temperature of a CSAT3 (Firmware v3)? You don't mention whether you determined the zero offset of the two CSAT3 before the experiment. Did you? The serial numbers of the CSAT3s tell us that they are relatively old instruments. How long ago was their last calibration? Figure 5: why the CSAT3 deviate that much although they should be better comparable. Could it be related to zero offsets or old calibration?

Technical corrections

Abstract

2/16    (Wieser et al., 2001). Full stop

3/33    indications of a

4/31    synchronized how? Please be more specific how this was done? Why it is that important if you compare just average quantities?

5/5    DE-Fen ?

6/5     It looks shaky. Were there guy wires?

7/2     All data were processed

7/15    DE-Fen

8/6     total wind velocity? You mean the magnitude of the 3d vector i.e. $=(u^2+v^2+w^2)^{0.5}$ compared to the horizontal wind speed $=(u^2+v^2)^{0.5}$, which is your mean wind speed?

12/5    was chosen as etalon for

12/15   CSAT3 is very good except for

14/5    Young.81000RE

15/5    The lower row is slightly too large so the y-axis is missing

16/3    of this study for many

16/7    etalon  because  (is a redundant, or omit "For this comparison")

17/7    error  due

17/6    measurement systems (?)

20/2    243-251 instead of 363-372

21/6    Frank and Massman listed twice

---

## Referee Comment (RC1) · J. Frank (Referee) · 26 Sep 2017

The comment was uploaded in the form of a supplement:
https://www.atmos-meas-tech-discuss.net/amt-2017-284/amt-2017-284-RC1-
supplement.pdf

---

## Author Comment (AC1) · 20 Nov 2017

**Final author response**

Journal: AMT Title: Field intercomparison of prevailing sonic anemometers Author(s): Matthias Mauder and Matthias J. Zeeman MS No.: amt-2017-284 MS Type: Research article

We are grateful for the very valuable and constructive comments by the reviewer. The comments are in black font, responses are presented in blue font and changes in the manuscript in red font.

**Response to RC1 (by John Frank)**

John Frank, USDA Forest Service In this manuscript seven sonic anemometers from six nonorthogonal designs and four manufactures are compared during a two and a half-week study at the TERENO/ICOS site in southern Germany. Half-hourly mean wind velocity, mean and standard deviation of temperature, standard deviation of vertical wind velocity, friction velocity, and buoyancy flux were compared between the seven anemometers. In general, all anemometers were reasonably similar, with the largest discrepancy being temperature measurements with the Gill anemometers.

The topic of this study is timely, with a growing interest in the accuracy in sonic anemometer measurements. The work presented here is clear, convincing, and thorough. I believe there is one main comment and a few minor issues that need to be addressed before it is acceptable for publication in Atmospheric Measurement Techniques.

I have one main comment that should be addressed. The discussion states "it is also possible that the flow distortion errors were very small for our experimental set-up because the angles of attack are close to being surface-parallel" with support from Figure 9 showing that most angles were within ±6° with a majority falling within an even narrower window. I am concerned that there is a high likelihood that each of these non-orthogonal anemometers have erroneous and unpredictable measurements of ow for winds with very small angles-of-attack. This is described in Appendix 2 of Frank et al. (2016a) where it is demonstrated for the CSAT3 that as the angle of-attack approaches 0° (i.e., near surface-parallel) that systematic w measurement uncertainties approach ±∞. This finding could be extended to any of the non-orthogonal anemometers included in this study, and I have included an Appendix at the end in this review to expand upon this topic. The discussion also states "The other interpretation that all anemometers are afflicted with the same bias appears less likely, since it is difficult to imagine that several instruments measure the same quantity equally wrong, despite the obvious differences in sensor geometry and internal data processing."

The figure and the text passages that this reviewer refers to are not part of the current manuscript published in AMTD. Based on the reviewer's comments in the Quick Review, we have double-checked our results and we found that there was a mistake in the calculation of the angles of attack. The spread in angles of attack during our experiment was actually much larger as previously thought, i.e. a standard deviation of 15° rather than  $\pm$ 6°. This standard deviation of 15° is at the upper end of values reported for previous intercomparison experiments. Therefore, we have also changed the line of arguments in the discussion in accordance with the new results. We now really believe that the different instruments all show the same biases despite their differences in geometry and internal corrections. Since there is strong evidence for a bias in  $\sigma_w$  of the CSAT3 from other studies (Horst et al. 2015, Frank et al. 2016, Huq et al. 2017), this seems to be the only logical conclusion.

Nevertheless, we highly appreciate the reviewer's extensive comments on the problem of nonorthogonal anemometers, which he provided in the Appendix of RC1.

I suggest that "the other interpretation" might actually be true, that these systematic w measurement uncertainties that approach  $\pm \infty$  could explain why these "instruments measure the same quantity equally wrong".

We now fully agree that these instruments measure the same quantity equally wrong. We have informed the editor about the changes that we have made after the Quick Review stage, but unfortunately, those notes did not reach the reviewer in time.

A key component of this experiment is the inclusion of the CSAT3, which does not apply any transducer shadowing correction, which has been modeled (Huq et al. 2017) and observed (Horst et al. 2015) to have transducer shadowing errors that when transformed into orthogonal coordinates lead to unpredictable measurements for near surfaceparallel winds (Frank et al. 2016a). It is nearly impossible to know exactly what happens for near surface-parallel winds, but a simple evaluation of the Kaimal correction (Kaimal et al. 1990) applied to the CSAT3 yields a range of  $\pm ~4^{\circ}$  where systematic measurement errors approach  $\pm \infty$  as shown in Figure 3f of Frank et al. (2016a). Because most angles were within  $\pm 6^{\circ}$  in this study, I consider it extremely likely that the CSAT3 data has such errors. At the same time, all other anemometers in this study are probably more susceptible to this problem because their transducers are all tilted closer to the horizontal plane (45° for the Gill, R. M. Young and Metek versus 60° for the CSAT3).

I have two suggestions that can help address this issue. First, the authors could do a sensitivity analysis to quantify the potential impact of transducer shadowing on the CSAT3 ow and buoyancy flux measurements. I would suggest using both the piecewise (Kaimal et al. 1990) and sinusoidal (Wyngaard and Zhang 1985) corrections presented in the following appendix. While it is important to note that there is no consensus that these corrections are accurate (two studies have shown that they might account for about half of the shadowing (Frank et al. 2016b, Huq et al. 2017)), this will help to evaluate the statement that "the flow distortion errors were very small".

Again, this is a quotation from an earlier version of the manuscript, which is not part of the manuscript that is under review of this open discussion and published in AMTD. In contrast to the earlier version, we do not believe anymore that the flow distortion errors were very small, but rather that all tested anemometers are afflicted with a very similar flow distortion error. Therefore, we have even further strengthened the following statement accordingly:

A common significant systematic error of all tested instruments is quite possible, as suggested by Frank et al. (2016).

However, we find that this suggestion is extremely interesting for follow-up work, but we believe this is out of the scope of this manuscript. We acknowledge the need for an angle of attack dependent correction, but this should probably include data from multiple sites with different surface and vegetation properties.

Second, in conjunction with the histogram in Figure 9, an analysis of the relative contribution of winds from each angle of attack bin to the total  $\sigma$ w and buoyancy flux measurements would be useful. Figure 9 currently shows that most data is in the bins -1° to 0° and 0° to 1°. How important are the measurements in these bins to the total  $\sigma$ w and buoyancy flux measurements reported in figures

5 and 8? What is the contribution of winds that exist within -4° to 4° (i.e., a range over which the Kaimal correction as applied to the CSAT3 could conceivably result in unpredictable measurements)?

I very much look forward to the authors' critical evaluation of this topic as I believe it will be an immense benefit to the research community to thoroughly discuss non-orthogonal wind measurements.

We belief this is a misunderstanding, which stems from the fact the reviewer refers to an earlier version, which was altered during the Quick Review stage. Since the old Figure 9 was erroneous and is not included in the manuscript under discussion, we do not believe there is a need for further clarification here.

I have a minor comment about interpreting results based on offset/bias versus slope differences. My impression is that the authors focus more on offset/bias differences and less about slope. One example is on page 12, lines 4-5 "the fluctuations of sonic temperature agree much better". When I compared Tables 3 to 5, my attention immediately focused on the slopes, which are not much different. The average absolute difference from 1.00 (i.e., an extremely simple metric to summarize the group differences) was 4.0% for Table 3 (i.e., 1.05, 0.97, 1.01, 1.05, 1.06, 1.04 -> +5%, -3%, +1%, +5%, +6%, +4% -> (5+3+1+5+6+4)/6 = 4%) and 3.5% for Table 5. Similarly, on page 12, lines 12-14 it is commented the high slope of 1.06 "might be a direct consequence of the almost equally high regression slope of 1.05". This is not surprising, because ideally, slope errors in measuring Ts should commute to slope errors in  $\sigma$ Ts.

We agree that for flux measurements, an error in the slope is more severe that an error in the bias, since mean is always subtracted when calculating a covariance. The first sentence quoted in this comment refers to the Gill instruments. We think then this is a fair statement, because they have quite large deviations of the mean sonic temperatures, but its standard deviation appears comparable. We have modified this sentence for clarification:

Despite the large discrepancies of the mean sonic temperature measurements of the Gill instruments, the fluctuations of sonic temperature agree much better

We have also made the second quoted sentence stronger, because there is indeed a direct relation between the slope of the mean and the slope of the standard deviation:

The METEK.uSonic.omni stands out because it has the highest regression slope of 1.06, which is a direct consequence of the almost equally high regression slope of 1.05 for the mean sonic temperature measurement.

Specific comments:

The introduction is excellent, and one of the better that I've read for sonic anemometer studies. What is the sampling rate of the sonic anemometers?

This information has been added to the manuscript.

The sampling rate was 20 Hz, except for the CSAT3\_2, which was sampled at 60 Hz, and the Gill\_HS, which was sampled at 10 Hz.

Figure 2: It would be good to mention either on the figure or in the caption that the model of anemometer corresponds to the same names listed in Table 1.

We agree and we have added that information in the caption of Figure 2.

they are presented from left to right in the same order as they are listed in **Fehler! Verweisquelle konnte nicht gefunden werden.**

Page 7, line 15: Are the obstructed wind directions based on 30-minute mean direction or some other metric?

They are indeed based on 30-minute mean wind directions. We have added this information.

These quantities were filtered for rain (during the respective half hour or the half hour before as recorded by a Vaisala WXT520 sensor of the nearby TERENO station DEFen), obstructed wind directions  $\varphi$  based on 30-minute averages (70°

Page 16, line 6: "error of due" should be "error due".

Thanks, we have removed the word "of"

Figure 9: Are the magnitudes of the numbers of y-axis correct? The number of occurrences seems extremely low for instantaneous 10 Hz or 20 Hz data. Or is this half-hour average angle of-attack? The number of occurrences seems similar to the number of half-hours. If this is the case, then all text that refers to "small ... angles of attack" (beginning with page 16, lines 25-26) must be revised to reflect the instantaneous angles experienced by the anemometers.

Indeed, the values were too low due to a mistake in the data analysis routine. We have therefore removed this figure already after the Quick Review because this whole line of argument saying that we had exceptionally low flow angles and therefore low systematic errors does not hold anymore. For completeness, we now provide the standard deviation of the flow angle, which is 15°, ranging at the upper end of values reported in the literature. Thanks to the reviewer's remark in the Quick Review, we were already able to correct this mistake.

Page 18, line 11-14: The discussion of "type A" and "type B" should occur earlier in the paper than the conclusion.

We have added the following sentence to the discussion section:

Now, all tested instruments are within the limits that Mauder et al. (2006) classified as type A, i.e. sonic anemometers suitable for fundamental turbulence research.

---

## Author Comment (AC2)

**Final author response**

Journal: AMT
Title: Field intercomparison of prevailing sonic anemometers
Author(s): Matthias Mauder and Matthias J. Zeeman MS No.: amt-2017-284 MS Type: Research article

We are grateful for the very valuable and constructive comments by the reviewer. The comments are in black font, responses are presented in blue font and changes in the manuscript in red font.

**Response to RC2 (anonymous)**

General Comments "As the last comprehensive intercomparison experiments were conducted more than 10 years ago, …" is the motivation for the authors do carry out a new intercomparison for prevailing sonic anemometers. They present the analysis and the results in a well-prepared manuscript in a straight and standard way. The sonic-user community will be eager to see how the different instrument types perform. Insofar it is worthwhile to publish their results and it is perfectly within the scope of AMT. There are however some points the authors should address.

The explanation/discussion of the much better agreement is not convincing. I don't understand what the consistent digital data acquisition has to do with the better agreement. Give an example. And demonstrate how your quality tests improve the agreement.

The quality tests of Mauder et al. (2013) are the basis for the quality control applied for this intercomparison. The effect of these tests is presented in that paper as well, also for the same site. In addition, we applied a filter for rain and obstructed wind directions. Our wind sector filtering implies the exclusion of all obstructions, including the tripod and the neighboring systems at a distance of 9 m. In addition, the filtering for rainy periods was critical to exclude implausible measurements. In the discussion section of revised version, we have provided some examples of the comparison statistics for the dataset after only the tests of Mauder et al. (2013) had been applied and also for the dataset after exclusion of obstructed wind sectors before the filtering for rainy periods was applied.

On top of that, the filtering for obstructed wind direction sectors and for rain, as described in section 2.2, was crucial to remove poor quality data. Both additional steps improved the agreement between instruments considerably. For $\sigma_w$, regression slopes ranged between 1.00 and 1.24 and intercepts were between −0.05 and 0.00 m s$^{-1}$ after processing according to Mauder et al. (2013). After filtering for obstructed wind direction, slopes ranged between 0.98 and 1.22 and intercepts remained between −0.05 and 0.00 m s$^{-1}$. As can be seen from the results (**Fehler! Verweisquelle konnte nicht gefunden werden.**), the overall agreement further improved after the filtering for rainy periods. Especially, some outliers of the CSAT3_2, which did not have the rain-guard meshes at the transducer heads, were rejected after this step. The effect of the data filtering on other quantities, such as Hs, was smaller. Here, the slopes ranged already only between 0.97 and 1.00 after processing according to Mauder et al. (2013), which did not change much further after filtering for obstructed wind directions and for rainy periods (**Fehler! Verweisquelle konnte nicht gefunden werden.**). This can be explained by the fact that the scheme of Mauder et al. (2013) is designed for quality control of fluxes and not necessarily standard deviations. It therefore much stricter for $H_s$ than for $\sigma_w$.

You also indicate that contributions in changes of firmware might have an influence. Say more about that. And finally you say that five instruments apply "some sort" of correction. There is more information about the corrections, they should not be treated as blackbox. Add the information where it is existiong. E.g. the calibration files for R3 and HS are available and can be applied later (at least it used to be like that). One has the possibility to sample uncalibrated data and apply the calibration afterwards. Did you do that? The HS and the CSAT3: how would they compare then? The sonics from Young and Metek seem to be black boxes but they allow to switch on and off a wake or head correction. You probably used the sonics always with the corrections on. Any idea how strong the corrections are? It irritates me that an instrument like the Young 81000 with a magic wake correction is so close to the other instruments. Insofar I can understand that you find the good agreement "somewhat surprising" and I can follow your conclusion that this is rather a conservative estimate because of special conditions (small variation in angles-of-attac. I guess you investigated the differences on azimuthal dependencies).

It is possible to operate the Gill instruments in an uncalibrated mode, but our intention was to compare the anemometers in the configuration recommended by the manufacturer. We do not have information about a calibration of the Gill instruments that can be applied during post-processing. We also used the METEK with the head correction turned on. As far as we know, the CSAT3 is the only one of the tested instruments that does not apply any internal correction at the sensor level. We do not have all details about the corrections applied by the different manufacturers. Therefore, the scope of this comparison was limited to a characterization of a typical configuration as it is applied by most users. We have added text providing details about the settings and firmware versions.

All other settings were left at the factory-recommended values, including flow-distortion corrections. The differences due to different firmware versions are quite well documented for the CSAT3. Accordingly to Burns et al. (2012), discrepancies between firmware versions 3 and 4 occur mostly for the sonic temperature measurement and they become significant for wind speeds larger than 8 m s$^{-1}$. During our field campaign, wind speeds were mostly lower than 5 m s$^{-1}$ (Figure 4). Therefore, we do not expect large errors. Nevertheless, we used the same firmware version (ver4) for both CSAT3.

We as authors share the reviewer's irritation about missing or incomplete information on the sensor-based corrections in the respective manuals or firmware documents by all manufacturers. Campbell Scientific certainly has the more transparent policies with respect to the internal processing routines.

The angle-of-attack figure disappeared. It was surprising to see that the deviations from horizontal were that small (mostly within ±6°). Now there is a standard deviation of 15°. What happened?

Based on the reviewer's comments in the Quick Review, we have double-checked our results and we found that there was a mistake in the calculation of the angles of attack. The spread in angles of attack during our experiment was actually much larger as previously thought, i.e. a standard deviation of 15° rather than ±6°. This standard deviation of 15° is at the upper end of values reported for previous intercomparison experiments. Therefore, we have also changed the line of arguments in the discussion in accordance with the new results. We now really believe that the different instruments all show the same biases despite their differences in geometry and internal corrections. Since there is strong evidence for a bias in $\sigma_w$ of the CSAT3 from other studies (Horst et al. 2015, Frank et al. 2016, Huq et al. 2017), this seems to be the only logical conclusion.

I do not see the advantage of using the PCA load in the first place, for deciding on an etalon. Choosing rather one instrument for all comparison is much more stringent and makes it easier to compare the instruments.

We wanted to avoid any subjectivity in choosing the reference instrument. If we had chosen the CSAT3 as etalon, as was done for previous intercomparsions, the presentation of the results would have been biased, perhaps favoring the CSAT3. Moreover, such a priori decision would contradict one of our main conclusions, that they all are equally suitable for flux measurements. Therefore, we would like to stick with the PCA-based decision on the etalon, because it allows us to compare the all results with the "best" estimate of a certain quantity.

Comparison plots are a bit monotone and do not transmit much information. Plotting rather differences to reference than sonic value versus sonic value gives an immediate impression on statistics. For a direct connection to the scatter plots the regression results should be placed in the plots. Special features can be highlighted in the text.

Thanks for this suggestion. We agree that there are other ways to present the results of such an intercomparison. However, we followed the style of previous studies, which have also used similar scatter plots (Dyer et al., 1982; Fratini and Mauder, 2014; Mauder et al., 2007; Tsvang et al., 1985) in order to maintain comparability. The monotone nature of the plots in our study, in comparison to plots in previous studies with much more scatter, is an actual result.

Comment on the speed-related temperature of a CSAT3 (Firmware v3)? You don't mention whether you determined the zero offset of the two CSAT3 before the experiment. Did you? The serial numbers of the CSAT3s tell us that they are relatively old instruments. How long ago was their last calibration? Figure 5: why the CSAT3 deviate that much although they should be better comparable. Could it be related to zero offsets or old calibration?

The older of the two CSAT3 (SN 0771) had been sent for re-calibration to the manufacturer in 2014. This one has firmware version 4t. The other one (SN 1791) still has its original manufacturer calibration; it was purchased in 2009. It has firmware version 4. We did not determine the zero offset before the experiment.

**Technical corrections**

Abstract 2/16 (Wieser et al., 2001). Full stop

Thanks, full stop has been added.

3/33 indications of a

This has been corrected.

4/31 synchronized how? Please be more specific how this was done? Why it is that important if you compare just average quantities?

Here, we wanted to stress the importance of digital data acquisition with precise clocks in order to attribute the correct time stamp to each data line. Therefore, more information on these details are provided in the revised version:

Data from all instruments were digitally recorded on synchronized single-board computers (BeagleBone Black, BeagleBoard.org Foundation, Oakland Twp, MI, USA), equipped with temperature-compensated clocks (Chronodot, Macetech LLC, Vancouver, WA, USA), using an event-driven protocol for recording data lines, implemented in the Python programming language. The digital recording minimizes the influence of data cable properties on signal quality and minimizes the impact of loss of resolution by conversion between analog and digital signals outside the scope of the sensor. Issues stemming from cable properties usually have a more apparent effect on digital than on analog signal transmissions. In case of a signal deterioration by oxidation of contacts or loosening cable connections, digitally transmitted data lines will start to show up in a corrupted format, while loss of signal resolution in analog transmission may go unnoticed for some time. Therefore, the potential for added uncertainty to the observations recorded by analog data transmission can in part be avoided by digital communications.

5/5 DE-Fen ? 6/5 It looks shaky. Were there guy wires?

The tripods did not have guy wires. Nevertheless, they are more stable than they might look like on the photograph because the legs are partially hidden within the grass canopy. We also checked the spectra of the wind velocities and found no visible deviations from the typical inertial sub-range behavior, which might indicate vibrations of the masts at distinct frequencies.

7/2 All data were processed

This has been corrected.

7/15 DE-Fen

The missing dash has been added.

8/6 total wind velocity? You mean the magnitude of the 3d vector i.e. $=(u2 +v2 +w2)0.5$ compared to the horizontal wind speed $=(u2 +v2)0.5$, which is your mean wind speed?

Since we have applied double rotation, the mean vertical wind velocity is zero and the mean cross-wind velocity is also zero. For clarification, we have now consistently used the term "mean total wind velocity".

12/5 was chosen as etalon for

This has been corrected.

12/15 CSAT3 is very good except for

This has been corrected.

14/5 Young.81000RE

This has been corrected.

15/5 The lower row is slightly too large so the y-axis is missing

Thanks, reviewer 1 has also noticed this, and we have recompiled the figure.

16/3 of this study for many

This has been corrected.

16/7 etalon for this quantity because (is a redundant, or omit "For this comparison")

This has been corrected.

17/7 error of due

This has been corrected.

17/6 measurements systems (?)

This has been corrected.

20/2 243-251 instead of 363-372

This has been corrected

21/6 Frank and Massman listed twice

Thanks for the careful check of the reference list. This mistake has been corrected.